Exploration of gray matter alterations and cognitive function impairment in adolescents with first-episode non-suicidal self-injury and the associations with self-injury characteristics

Yu Rui 1
Liao Kaike 1
Chen Yuwei 1
Chen Xinyue 1
Peng Shiji 1
Li Xianfu 2
Liu Nian liunian@nsmc.edu.cn 1 3
1 Department of Radiology, Affiliated Hospital of North Sichuan Medical College , Nanchong City , Sichuan Province , China
2 Department of Radiation Oncology, Affiliated Hospital of North Sichuan Medical College , Nanchong City , Sichuan Province , China
3 Functional and Molecular Imaging Key Laboratory of Sichuan Province , Chengdu City , Sichuan Province , China
Liu Feng
Electronic publication date: 2025 Aug 26
Publication date: 2025
Volume: 13
Electronic Location ID: e19914
Received 2024 Nov 18; Accepted 2025 Jul 23
Copyright: ©2025 Yu et al.
Copyright year: 2025
Copyright holder: Yu et al.
License: This is an open access article distributed under the terms of the Creative Commons Attribution License, which permits unrestricted use, distribution, reproduction and adaptation in any medium and for any purpose provided that it is properly attributed. For attribution, the original author(s), title, publication source (PeerJ) and either DOI or URL of the article must be cited.
License URL: https://creativecommons.org/licenses/by/4.0/

Keywords: Non-suicidal self-injury, Magnetic resonance imaging, Gray matter volume, Cognitive functions, Self-injury characteristics

Funding: Sichuan Science and Technology Program 2024ZYD0272 Bureau of Science & Technology and Intellectual Property Nanchong City 22SXQT0305 Opening Project of Functional and Molecular Imaging Key Laboratory of Sichuan Province SCU-HM-202307001 This work was supported by the Sichuan Science and Technology Program (2024ZYD0272), the Bureau of Science & Technology and Intellectual Property Nanchong City (NO. 22SXQT0305), and the Opening Project of Functional and Molecular Imaging Key Laboratory of Sichuan Province (NO. SCU-HM-202307001). The funders had no role in study design, data collection and analysis, decision to publish, or preparation of the manuscript.

==============================
Background

It remains unclear if there are potentially associated abnormalities in gray matter (GM) and cognitive function in adolescents with non-suicidal self-injury (NSSI), and if there are associations with self-injury characteristics. Therefore, exploring the alterations in GM and cognitive functions and their associations with self-injury characteristics in adolescents with first-episode NSSI can provide imaging and clinical evidence for understanding the pathogenesis of NSSI.

Methods

In this cross-sectional study, we prospectively collected 29 adolescents (NSSI group) with first-episode NSSI and 28 healthy controls (HC group). Participants were scanned using a 3.0T MRI scanner. GM measures were extracted and compared between the NSSI group and the HC group using covariance analysis with total intracranial volume, age, sex, and years of education as covariates. Evaluate the cognitive functions of two groups and perform covariance analysis with years of education, age, and sex as covariates. The assessment of self-injury function was conducted using the Beck Scale for Suicide Ideation and the Ottawa Self-Injury Inventory. With years of education as the control variable, partial correlation analysis is carried out between GM volume (GMV) and cognitive functions. A mediation effect analysis was conducted on GMV, cognitive function, and NSSI to explore the relationships among them.

Results

The cognitive functions of the NSSI group are poorer than those of the HC group. Compared with the HC group, the NSSI group had decreased GMV in the left putamen and left nucleus accumbens and an increased GMV in the left rostral anterior cingulate cortex. In the NSSI group, the self-injury characteristics and poorer cognitive function are associated with abnormal alternations in GMV, and the poorer cognitive functions are also associated with the self-injury characteristics. The mediation analysis showed that the volume of the left rostral anterior cingulate cortex played a partial mediating role in the relationship between NSSI behavior and cognitive decline.

Background

Non-suicidal self-injury (NSSI) is defined as the direct and intentional destruction of one’s own body without lethal intent (Nock, 2010). The population at high risk for NSSI is adolescents, with the global incidence of NSSI in this group ranging approximately from 7.5% to 46.5% (Cipriano, Cella & Cotrufo, 2017). NSSI often starts in early adolescence and peaks in mid-adolescence, which can predict the tendency of suicide in adolescents (Plener et al., 2015; Kiekens et al., 2018). Understanding the neural pathogenesis of NSSI offers strategies for suicide prevention. A recent study showed that since the new coronavirus outbreak, the incidence of NSSI in adolescents has increased and has attracted increased amounts of attention (Zhang et al., 2020). In 2013, NSSI was recognized as a separate clinical disorder in the Diagnostic and Statistical Manual of Mental Disorders (DSM-V) (Zetterqvist, 2015), signifying that NSSI, as a significant psychological issue among adolescents, has garnered wide public concern. However, the pathogenesis of NSSI is still unclear. Therefore, studying the pathogenesis of first-episode NSSI is necessary.

Self-injury characteristics, which is a crucial aspect related to NSSI, pertains to the diverse underlying motives, purposes, or psychological and social needs that prompt individuals to engage in such behaviors (Taylor et al., 2018; Edmondson, Brennan & House, 2016). It can be generally categorized into intrapersonal and interpersonal functions. Intrapersonal functions are mainly associated with the individual’s internal psychological state, including internal emotion regulation, sensation seeking, addictiveness, and impulsivity. Interpersonal functions, on the other hand, involve the individual’s interaction with the external environment, such as external emotion regulation and social influence. These functions play an important role in understanding the complexity of NSSI as they may have implications for the manifestation and severity of the behavior (Andover & Morris, 2014; Taylor et al., 2018).

Cognitive functions play an important role in the risk of NSSI. Previous studies have found that NSSI patients show cognitive deficits in reasoning and problem-solving, inhibitory control, executive function, and working memory (Hu et al., 2021; Mürner-Lavanchy et al., 2022; Wang et al., 2023). A large number of studies have confirmed that abnormalities in individual cognitive functions, including motor speed, verbal memory, visual memory, cognitive flexibility, working memory, etc., are often accompanied by abnormalities in brain function or structure (Zhao et al., 2023; Xiao et al., 2024; Jing et al., 2024). Lu et al. (2024) have explored the brain functional connectivity and cognitive functions of NSSI patients and found that the abnormal functional connectivity of the frontotemporal cortex in NSSI patients is positively correlated with poorer cognitive functions such as language learning and working memory. Dahlgren et al. (2018) also found alterations in neural activation patterns in the cingulate cortex and prefrontal cortex during cognitive interference tasks in individuals with NSSI who were assigned female sex at birth (AFAB). These studies indicate that in NSSI patients, alterations in brain function are closely related to cognitive functions such as working memory, processing speed, executive function, and verbal memory. Considering gray matter as a core indicator of brain structure, which directly reflects neuronal density, synaptic connection strength, and the functional integrity of brain regions, chronic emotional dysregulation has been demonstrated to induce gray matter alterations in brain regions through neuroplasticity mechanisms (Zhao et al., 2023; Xiao et al., 2024). These gray matter alterations are closely associated with disruptions in neural network connectivity, potentially leading to impairments in cognitive abilities such as working memory and executive function (Eres et al., 2015; Killgore et al., 2016; Maat et al., 2016). Therefore, gray matter alterations may mediate the association between NSSI and cognitive decline. From a temporal-mechanical perspective, chronic emotional dysregulation in NSSI likely precedes gray matter volume (GMV) alterations. Neuroplasticity studies show that prolonged stress induces dendritic pruning in the nucleus accumbens, while the rostral anterior cingulate cortex (rACC) may exhibit compensatory hypertrophy in early-stage emotional dysregulation (Eres et al., 2015; Killgore et al., 2016). These structural changes then disrupt cortico-striatal circuit connectivity, leading to subsequent cognitive impairments in working memory and motor control (Maat et al., 2016). Although previous studies have demonstrated an association between impaired cognitive function and functional magnetic resonance imaging (MRI) alterations in NSSI patients, the relationship between gray matter alterations and impaired cognitive function in NSSI adolescents remains unclear. Therefore, it is necessary to investigate the alterations in gray matter and cognitive function in adolescents with first-episode NSSI and their interrelationships with self-injury characteristics.

In this study, we investigated the alterations in gray matter (including cortical volume, cortical thickness, cortical surface area, and subcortical volume) and cognitive functions between adolescents with first-episode NSSI and healthy controls (HC), as well as their relationships with self-injury characteristics. Based on this, we hypothesize that adolescents with first-episode NSSI will exhibit significant gray matter alterations in nucleus accumbens, putamen, and rACC, and have cognitive impairments in working memory, motor speed, and verbal fluency. We further propose that GMV alterations in these regions mediate the association between NSSI and cognitive deficits.

Methods

Participants

This is a cross-sectional study which was approved by the Ethics Committee of the Affiliated Hospital of North Sichuan Medical College (2022ER406-1), and written informed consent was obtained from each participant. This study prospectively collected adolescent patients with first-episode NSSI from the Affiliated Hospital of North Sichuan Medical College from November 2022 to July 2023.

The criteria for inclusion in the NSSI group were as follows: (1) a first diagnosis within the NSSI in accordance with the DSM-V; (2) Han nationality, right-handed; (3) had no history of head trauma; and (4) were not treated with psychotropic drugs.

The inclusion criteria for individuals in the HC group were as follows: (1) confirmed via structured interviews with participants and guardians to have no history of neuropsychiatric illness after DSM-V nonpatient version evaluation and screening; (2) Han nationality, right-handed; (3) lacked a history of head trauma; and (4) lacked a history of medication within 30 days before the examination.

The exclusion criteria for the two groups were as follows: (1) contraindication for MRI examination; (2) suffering from serious diseases of the central nervous system (such as brain tumors, epilepsy, multiple sclerosis, etc.), severe body diseases (such as hepatitis, etc.), or severe drug allergy; (3) recently had a severe infection, surgical history, or a history of severe head trauma; (4) stereotypical dyskinesia; and (5) alcohol or drug dependence.

Finally, thirty-five adolescents with NSSI were recruited, among whom six were excluded because they did not meet the inclusion criteria (one had contraindications for MRI, one had taken antidepressant drugs, and four had other mental disorders). A total of 29 patients with NSSI were enrolled (the NSSI group)—27 AFABs and two males—with a median age of 14.00 years (range: 13–19 years). HC were recruited from the same or adjacent community as the NSSI patients, and all the participants had similar socioeconomic and educational backgrounds. Thirty-seven HC were recruited, among whom nine were excluded because they did not meet the inclusion criteria (two had contraindications for MRI, three had a history of medication within 30 days before the examination, and four had a history of neuropsychiatric illness). A total of 28 HC were enrolled (the HC group), including 21 AFABs and seven males (supplementary Figure). The HC had a median age of 15.00 years (range: 14–16 years). The diagnosis of NSSI was determined by a psychiatrist using the Structured Interview for the DSM-V. All the participants were imaged using the same scanner. Motion artifacts and scan quality were evaluated via visual inspection and quantitative metrics (e.g., framewise displacement), with no participants excluded for imaging quality issues.

Image data acquisition

All magnetic resonance imaging (MRI) data were obtained by a 3.0T magnetic resonance scanner (Discovery 750; GE Healthcare, Milwaukee, WI, USA) equipped with a 32-channel phased array head coil. All participants used tight but comfortable foam pads to reduce head movement and earplugs to reduce scanning noise. All participants underwent a uniform protocol, including axial high-resolution T1-weighted imaging with a volumetric three-dimensional brain volume imaging sequence (BRAVO). The specific parameters used were as follows: matrix=256×256, repetition time = 8.5 ms, echo time = 2.0 ms, flip angle = 12°, field of view = 256 × 256 mm2, voxel size = 1 mm × 1 mm × 1 mm, and axial slices = 188.

Clinical data acquisition

Clinical data were collected and assessed on the day of the MRI examination. The demographic data included name, sex, age, years of education, etc. The clinical characteristics included the Beck Scale for Suicide Ideation (BSSI), the Brief Assessment of Cognition in Schizophrenia (BACS), and the Ottawa Self-Injury Inventory (OSI) (Beck, Kovacs & Weissman, 1979; Keefe et al., 2004; Martin et al., 2013b). The BSSI has a maximum total score of 100, with higher scores indicating a greater risk of suicide. The BACS includes six factors: verbal memory, working memory, motor speed, verbal fluency, attention and speed of information processing, and executive functions. Meanwhile, the total score of the BACS scale, which reflects the overall cognitive functions, was calculated as well. For all factors, lower scores indicate poorer cognitive functioning (Hidese et al., 2018). The OSI scale is used to evaluate the self-injury characteristics of NSSI adolescents and includes five factors: internal emotion regulation, social influence, external emotion regulation, sensation seeking, and addiction characteristics. For the five factors, the higher the score for each factor, the more severe the clinical symptoms were (Luo et al., 2024). For participants under 18 years of age, diagnosis and assessment were performed by consensus between themselves and their parents or guardians. We exerted strenuous efforts to ensure that the participants completed all the assessments. However, the OSI data of seven subjects in the NSSI group remained unavailable, thus they were excluded from the subsequent OSI-related analyses to ensure the reliability of the research results.

Imaging processing

The completely automated and verified segmentation software FreeSurfer (version 6.0, https://surfer.nmr.mgh.harvard.edu/) was adopted for cortical modeling, volumetric segmentation, and for measuring cortical volume, cortical thickness, cortical surface area, and subcortical volume. This was achieved by using image intensities and continuity information from the entire MR volume to construct representations of the gray/white matter boundary and pial surface (Dale, Fischl & Sereno, 1999; Ségonne, Pacheco & Fischl, 2007; Fischl, Liu & Dale, 2001). The processing procedures, which encompassed skull stripping, segmentation of gray matter, white matter, and cerebrospinal fluid, cortical surface reconstruction, normalization, and parcellation, were initiated using shared information from the within-subject template (Fischl, Liu & Dale, 2001; Ségonne, Pacheco & Fischl, 2007; Reuter et al., 2012). The aforementioned procedure produced average cortical volumes, cortical thickness, and cortical surface areas for 68 regions and subcortical volumes for 14 regions of interest (nucleus accumbens, amygdala, caudate, hippocampus, pallidum, putamen, and thalamus in each hemisphere) by employing the Desikan-Killiany atlas template (Desikan et al., 2006). Postprocessing visual inspection regarding the quality of the imaging process was carried out without awareness of subject characteristics.

Statistical analysis

The Kolmogorov–Smirnov test was used to assess normality, and Levene’s test was employed to evaluate homogeneity of variance. For variables failing to meet these assumptions (such as social influence and sensation seeking), data transformation methods were applied to enhance normality and stabilize variances. The Mann–Whitney U test was used to compare the differences between the two groups in age and years of education. Sex differences were compared by Fisher’s exact test. Covariance analysis was carried out on the cortical volume, cortical thickness, and surface area of all 68 brain regions, as well as the subcortical volume of 14 brain regions, with estimated total intracranial volume, age, sex, and years of education as covariates. The scores of the BACS scale were also analyzed by covariance analysis with age, sex, and years of education as covariates. Multiple comparisons for each of the 68 cortical measurements and 14 subcortical volumes were controlled using a false discovery rate criterion. Partial correlation analysis on gray matter, cognitive function, and self-injury characteristics showing significant differences is carried out with years of education as a control variable. Specifically, the correlation analyses included: (1) the relationship between alterations in gray matter indices and cognitive decline; (2) the relationship between alterations in gray matter indices and self-injury characteristics; and (3) the relationship between cognitive decline and self-injury characteristics. The significance level of the data was set at p = 0.05 (two-tailed).

Mediation model analyses

To explore the mediation effect of gray matter alterations on the relationship between NSSI and cognitive function, a mediation model analysis was carried out. Whether an individual engaged in NSSI was designated as the independent variable (X). The gray matter structure of brain regions with significant alterations was set as the mediator variable (M). And the cognitive function scores that showed significant differences between the two groups were used as the dependent variable (Y). A bias-corrected bootstrapped mediation analysis was performed using the SPSS macro PROCESS version 4.1 to test the indirect effect. The point estimate of the indirect effect was considered statistically significant if the 95% confidence interval obtained by bootstrapping (with 5,000 iterations) did not include zero.

Results

Differences in demographic and clinical characteristics between the NSSI group and the HC group

The demographic and clinical characteristics of the study participants are shown in Table 1. There were no significant differences in age, sex, or years of education between the NSSI and HC groups. The total BACS score, working memory score, motor speed score, and verbal fluency score in the NSSI group were lower than those in the HC group (p < 0.05).

Table 1 Demographical characteristics of the participants.

Demographic/clinical	NSSI	HC	t/Z/F statistic	P value	ηp2	
Sex (female/male)	27/2	21/7	–	0.131	–	
Agea	14.00 (2.00)	15.00 (0.00)	−1.599	0.110	–	
Years of educationa	8.00 (2.00)	9.00 (1.00)	−1.810	0.070	–	
BSSIb	60.14 ± 13.07	–	–	–	–	
OSI						
Internal emotion regulationb	15.68 ± 5.15	–	–	–	–	
Social influencea	6.14 (2.50)	–	–	–	–	
External emotion regulationb	7.59 ± 3.46	–	–	–	–	
Sensation seekinga	4.86 (3.00)	–	–	–	–	
Addiction characteristicsb	12.09 ± 5.22	–	–	–	–	
BACS						
Verbal memoryb	38.34 ± 18.91	43.04 ± 7.71	1.356	0.193	0.032	
Working memoryb	24.28 ± 4.65	28.00 ± 0.00	17.172	<0.001*	0.256	
Motor speedb	77.72 ± 9.35	88.00 ± 0.00	59.395	<0.001*	0.516	
Verbal fluencyb	23.41 ± 4.75	28.25 ± 5.91	11.608	0.004*	0.146	
Attention and speed of information processingb	37.17 ± 7.15	39.64 ± 4.89	2.392	0.160	0.038	
Executive functionsb	17.07 ± 3.00	17.64 ± 2.36	0.551	0.598	0.005	
Total scoreb	218.00 ± 27.14	244.57 ± 11.09	23.809	<0.001*	0.305	
Notes.

For all the test values shown in this table, the degrees of freedom is 54. NSSI, Non Suicidal Self-Injury; HC, Healthy Control; BSSI, Beck Scale for Suicide Ideation; OSI, Ottawa Self-injury Inventory; BACS, Brief Assessment of Cognition in Schizophrenia

a Data represented median (interquartile range).

b Data represented mean ± standard deviation.

* p < 0.05.

Differences in gray matter structure between the NSSI group and the HC group

Compared with the HC group, the NSSI group had decreased GMV in the left putamen and left nucleus accumbens and increased GMV in the left rACC (p < 0.05, corrected; Fig. 1; Table 2). Between the two groups, there is no significant difference in GMV of the remaining brain regions except for those mentioned above. Additionally, there is no significant difference in the cortical thickness and surface area of all regions (Supplementary Table).

Figure 1 Gray matter volume differences between the NSSI group and HC group.

Greater gray matter volume in adolescents with NSSI than in healthy controls are indicated by red/warm color, and lower volume are indicated by blue/cold color. L, left hemisphere; R, right hemisphere; rACC.L, Left rostral Anterior Cingulate Cortex; Put.L, Left Putamen; NAc.L, Left Nucleus Accumbens.

Table 2 Gray matter volume differences between NSSI group and HC group.

Region	NSSI (mm3)	HC (mm3)	F statistic	FDR corrected q statistic	ηp2	
L-Put	4,880.40 ± 410.76	5,310.08 ± 525.93	7.558	0.028	0.129	
L-NAc	285.88 ± 56.80	338.13 ± 63.73	8.179	0.028	0.138	
L-rACC	2,693.79 ± 584.117	2,299.57 ± 636.78	12.368	0.034	0.195	
Notes.

For all the test values shown in this table, the degrees of freedom is 54. The data in this table are represented as “mean ± standard deviation.” NSSI, Non-Suicidal Self- Injury; HC, Healthy Control; L-Put, Left-Putamen; L-NAc, Left Nucleus Accumbens; L-rACC, Left rostral Anterior Cingulate Cortex; FDR, False Discovery Rate.

Relationship among abnormal GMV, cognitive function, and self-injury characteristics

The GMV of the left nucleus accumbens shows a significant negative correlation with motor speed in cognitive function (r =  − 0.438, p < 0.05, Fig. 2A) and social influence in self-injury characteristics (r =  − 0.476, p < 0.05, Fig. 2B). The GMV of the left rACC showed significant positive correlations with working memory (r = 0.440, p < 0.05) and motor speed (r = 0.499, p < 0.05, Fig. 2C) in cognitive function. In the correlation between cognitive functions and self-injury characteristics, verbal fluency shows a negative correlation with social influence (r =  − 0.469, p < 0.05, Fig. 2D). The overall cognitive function also shows a negative correlation with external emotion regulation (r =  − 0.438, p < 0.05, Fig. 2E) and sensation seeking (r =  − 0.442, p < 0.05, Fig. 2F). There is no significant correlation between the GMV of the left putamen and clinical characteristics. The above results indicate that in the NSSI group, the greater GMV of the left nucleus accumbens or the smaller GMV of the left rACC, the more severe the self-injury characteristics and the worse the cognitive function. Moreover, the correlation analysis between the self-injury characteristics and cognitive function also shows that the more severe the self-injury characteristics is, the worse the cognitive function becomes.

Figure 2 Correlation among gray matter volume, cognitive function and self-injury characteristics.

Participants with missing data on the OSI scale were excluded from the calculation of degrees of freedom in all statistical analyses related to the OSI scale. For the test values shown in A and C, the degrees of freedom are 54. For the test values presented in B, D, E, and F, the degrees of freedom are 48. (A) The correlation between the volume of the left nucleus accumbens and motor speed; (B) The correlation between the volume of the left nucleus accumbens and social influence; (C) The correlation between the volume of the left rostral anterior cingulate cortex and motor speed as well as working memory; (D) The correlation between the verbal fluency and the social influence; (E) The correlation between the total score of the Brief Assessment of Cognition in Schizophrenia scale and the external emotion regulation; (F) The correlation between the total score of the Brief Assessment of Cognition in Schizophrenia scale and the sensation seeking.

Mediation effect of GMV on the relationship between NSSI and cognitive function

The mediation effect analysis indicated that the volume of the left rACC played a partial mediating role in the relationship between NSSI and cognitive decline (a =  − 0.619, p < 0.05; b = 0.233, p < 0.05; c′ = 1.368, p < 0.001; c = 1.224, p < 0.001; indirect effects = −1.209, 95% CI [−3.070 to −0.068]) (Fig. 3). When the left putamen and nucleus accumbens were used as independent variables for the same mediation analysis, no results with statistical significance were obtained.

Figure 3 Mediation effect of GMV on the relationship between NSSI and cognitive function.

Mediation analysis was performed using bias-corrected bootstrap resampling (5,000 iterations, n = 57). Path labels: c = total effect of NSSI on cognitive function; c′ = direct effect of NSSI on cognitive function after adjusting for mediator; a = effect of NSSI on left rACC volume; b = effect of left rACC volume on cognitive function. β coefficients shown; *p < 0.05, ***p < 0.001. rACC.L, left rostral anterior cingulate cortex; NSSI, non-suicidal self-injury.

Discussion

The study is intended to explore alterations in gray matter and cognitive function in adolescents with first-episode NSSI and their interrelationships with self-injury characteristics.

We found that compared with the HC group, the NSSI group showed reduced GMV in the putamen and nucleus accumbens, and increased GMV in the rACC. These alterations may be related to cognitive function and self-injurious behavior. NSSI adolescents demonstrated significantly poorer cognitive functions compared to HC. And the abnormalities in cognitive functions are significantly correlated with both GMV and the self-injury characteristics. Additionally, the alterations in GMV play a crucial mediating role in the influence of NSSI on adolescents’ cognitive functions. These findings suggest that structural abnormalities in the rACC, nucleus accumbens, and putamen, coupled with cognitive impairments, may serve as potential neurobiological markers for NSSI in adolescents. Early neuroimaging and cognitive assessments targeting these brain regions and functions could aid in the early identification of youth at risk for NSSI, informing the development of targeted interventions to improve emotional regulation and cognitive control.

Differences in GMV between the NSSI group and the HC group

One of the main findings of this study is that there is a decrease in GMV in the left nucleus accumbens and left putamen of adolescents with NSSI. Our results support the findings of prior MRI research regarding NSSI (Cullen et al., 2020; Chen et al., 2023a; Yi et al., 2024). Yi et al. (2024) discovered that there was a decreased volume and altered functional connectivity in the putamen of NSSI patients, and it was related to the addiction characteristics in the self-injury characteristics. Similarly, Chen et al. (2023a) also found that there were abnormal functional connections between the nucleus accumbens and the putamen in NSSI adolescents, and the abnormal functional connections of the putamen were also related to the addiction characteristics. The nucleus accumbens is the core of the reward circuits, and the putamen is also an important component of the reward circuits. They play a major role in reward perception, decision-making, and behavioural control (Haber & Knutson, 2010; Cullen et al., 2020). Neurons in the putamen can receive input signals from other regions and transmit them to related areas such as the nucleus accumbens. When rewarding stimuli are received, dopaminergic neurons in the nucleus accumbens are activated to release dopamine, resulting in a sense of pleasure and reward and stimulating behavioural responses to seek more rewards (Cullen et al., 2020). Abnormal reductions in GMV within the nucleus accumbens and putamen are associated with deviations in reward perception and seeking, thereby increasing the risk of self-injurious behaviour (Tsypes et al., 2018; Burke et al., 2022). The decreased GMV in the nucleus accumbens was significantly negatively correlated with self-injury characteristics, further supporting the association between altered GMV and NSSI.

Another important finding of this study is that there is an increase in GMV in the rACC of the NSSI group. Notably, this finding contrasts with prior reports of rACC volume reductions in AFAB adolescents with NSSI (Ando et al., 2018). A plausible explanation is that early-stage NSSI elicits compensatory neuroplastic alterations in rACC, attempting to maintain emotional regulation. However, this hypothesis requires validation, as cross-sectional designs cannot resolve the temporal sequence of structural-functional alterations. As an important part of the limbic system, the rACC plays an important role in emotional and cognitive processes (Rolls et al., 2022; Sun et al., 2023; Wei et al., 2024; Liao et al., 2024). In terms of emotional regulation, the rACC participates in the assessment, recognition, and regulation of emotions. It can quickly respond to emotional stimuli and plays a central role in resolving emotional conflicts (Ochsner et al., 2009; Ray & Zald, 2012). Abnormalities in the rACC of adolescents with NSSI are associated with impaired emotional regulation function, and result in an unstable emotional state and increased distress. Then, this obstacle in emotional regulation may prompt individuals to resort to extreme means to cope with emotional stress and thereby engage in NSSI (Etkin, Egner & Kalisch, 2011; Shackman et al., 2011; In-Albon & Schmid, 2012; In-Albon et al., 2013). Notably, Alarcon et al. (2019) discovered a significant association between rACC abnormalities in adolescents and both suicidal ideation and suicide attempts. Given that adolescents with NSSI also display rACC aberrations, it suggests that NSSI likely disrupts the normal function of the rACC. This disruption, in turn, impairs emotional regulation and cognitive control in the adolescent brain, thus increasing the adolescent suicide risk. The persistent accumulation of these risks has a high likelihood of culminating in the transformation of NSSI behaviors into suicidal behaviors (Muehlenkamp & Brausch, 2019). From a cognitive function perspective, as one of the important regions for cognitive control function, the rACC is closely related to the planning and execution of goal-directed behaviors and plays a crucial role in the allocation and adjustment of attention (Devinsky, Morrell & Vogt, 1995; Cruz et al., 2023). Altered GMV in the rACC may interfere with its normal cognitive control function, affecting an individual’s ability to plan and execute tasks and effectively regulate attention, which is associated with an inability to accurately assess behavioral consequences and risks during decision-making and a higher probability of NSSI (Botvinick, 2007; Shackman et al., 2011; Hiser & Koenigs, 2018; Rolls, 2023). The cognitive functions assessed in this study were significantly lower in adolescents with NSSI compared to HC. The increased GMV of the rACC shows a significant positive correlation with cognitive functions such as working memory and motor speed, which further indicates that the abnormality of the rACC is closely related to cognitive functions in NSSI.

Differences in cognitive functions between the NSSI group and the HC group

The results of this study demonstrate that the NSSI group exhibited poorer cognitive functions than the HC group, specifically evidenced by significant impairments in working memory, motor speed, and verbal fluency. Previous studies have confirmed cognitive processing abnormalities, such as executive function and working memory impairments, in NSSI adolescents (Chen et al., 2023b; Duncan-Plummer et al., 2023). This study further reveals that their cognitive impairments involve overall cognitive ability, motor speed, and verbal fluency, providing novel evidence for cognitive function research in this population. Mechanistically, cognitive functions are implicated in the pathogenesis of NSSI by influencing learning and memory, decision-making, and behavioral regulation (Martin et al., 2013a; Del Missier et al., 2013; Tournikioti et al., 2022). Through conducting multi-unit analyses on NSSI adolescents, Başgöze et al. (2023) found that significant cognitive function abnormalities in NSSI patients were associated with NSSI severity. Specifically, cognitive impairment may render individuals unable to evaluate the harms of self-injury, impair their ability to select adaptive coping strategies under emotional stress, and thereby be associated with an increased likelihood of alleviating distress through NSSI (Nilsson et al., 2021; Park & Ammerman, 2023). This cognitive-behavioral pattern is associated with both the onset of NSSI and an elevated recurrence risk.

Relationships among gray matter structure, cognitive function, and NSSI

In this study, for the NSSI group, the altered GMV of the nucleus accumbens and rACC shows a significant correlation with cognitive functions including working memory and motor speed and self-injury characteristics of social influence. This indicates that the alterations in GMV of NSSI adolescents are closely related to the impairment of cognitive functions and self-injury characteristics. Interestingly, we also found that self-injury characteristics, including social influence, external emotion regulation, and sensation seeking, show a significant negative correlation with impaired cognitive functions. This further proves that there is a complex and close relationship between NSSI and cognitive functions.

Mediation effect of GMV alterations between NSSI and cognitive function

The results of the mediation analysis show that NSSI significantly influences both the impairment of cognitive function and the alteration of rACC volume. Additionally, the alteration of rACC volume plays a partial mediating role in the process by which NSSI affects adolescents’ cognitive function. And specifically, NSSI directly affects the impairment of cognitive function and further impairs cognitive function by influencing the volume of rACC. This finding aligns with previous research highlighting the role of rACC in emotional regulation and cognitive processing, further bolstering the evidence for the influence mechanism of NSSI on cognitive functions. However, cross-sectional study limits causal inference, as it cannot determine temporal precedence or rule out reverse causality. Thus, longitudinal studies are needed to validate this causal pathway.

Prospective insights into NSSI studies

Future research should explore these associations using more advanced imaging techniques such as 7T T1-structural scans. These scans are capable of optimizing the quantification of gray matter alterations. The optimized quantification of GMV alterations by these scans further clarifies the complex relationships among GMV, cognitive functions, and NSSI, and reveals more previously unnoticed details. Additionally, combining different imaging modalities such as functional MRI and diffusion tensor imaging can provide a comprehensive depiction of the neural networks involved in NSSI. Through the integration of functional and structural information, insights can be obtained into how GMV alterations affect cognitive and emotional networks, as well as how these alterations interact with NSSI. This will provide a more comprehensive understanding of the intricate interactions among gray matter structure, cognitive functions, and NSSI, potentially uncovering new mechanisms not yet discovered in this study.

Limitations

Despite the valuable insights of this study, several limitations need to be acknowledged. Firstly, the small sample size and skewed gender distribution precluded an assessment of NSSI severity among participants, limiting the generalizability of findings to male adolescents. Larger samples with graded comparisons of NSSI severity are needed to facilitate more comprehensive NSSI research. Second, clinical characteristics of some participants were obtained via self-rating scales, potentially introducing self-report bias. Thus, the influence of participants’ subjective factors on results must be considered. Third, potential biases and confounding variables related to psychosocial factors (e.g., family environment, trauma history) were not fully explored. Future research should deeply evaluate these psychosocial factors to enhance understanding of NSSI’s pathophysiological mechanisms. Additionally, all participants were treatment-naive, which may restrict the applicability of results to adolescents with prior psychiatric treatment.

Conclusion

In conclusion, the results of this study indicate that there are alterations in GMV, including the nucleus accumbens, putamen, and rACC, in adolescents with first-episode NSSI, which are associated with self-injury characteristics and cognitive functions. Moreover, the impaired cognitive functions are also associated with self-injury characteristics. These findings reveal a complex and close relationship among GMV, cognitive function, and self-injury characteristics in adolescents with NSSI. These findings contribute to the understanding of the neural mechanisms underlying NSSI.

Supplemental Information

Supplemental Information 1 Raw data

Supplemental Information 2 STROBE checklist

Supplemental Information 3 Flowchart of Study Participants

Supplemental Information 4 Differences in Gray Matter Indices between NSSI Group and HC Group

NSSI, non-suicidal self-injury; HC, healthy control.

Supplemental Information 5 Codebook

Special thanks to the adolescents and families for their time and effort.

Additional Information and Declarations

Competing Interests

Author Contributions

Human Ethics

Data Availability

The authors declare there are no competing interests.

Rui Yu conceived and designed the experiments, performed the experiments, analyzed the data, prepared figures and/or tables, authored or reviewed drafts of the article, and approved the final draft.

Kaike Liao performed the experiments, authored or reviewed drafts of the article, and approved the final draft.

Yuwei Chen performed the experiments, authored or reviewed drafts of the article, and approved the final draft.

Xinyue Chen performed the experiments, authored or reviewed drafts of the article, and approved the final draft.

Shiji Peng performed the experiments, authored or reviewed drafts of the article, and approved the final draft.

Xianfu Li performed the experiments, authored or reviewed drafts of the article, and approved the final draft.

Nian Liu conceived and designed the experiments, performed the experiments, authored or reviewed drafts of the article, and approved the final draft.

The following information was supplied relating to ethical approvals (i.e., approving body and any reference numbers):

The Ethics Approval Committee of Affiliated Hospital of North Sichuan Medical College approval to perform the study within its facilities (Ethical Application Ref: 2022ER406-1).

The following information was supplied regarding data availability:

Raw data is available in the Supplemental Files.

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
