# Peer review of "Exploration of gray matter alterations and cognitive function impairment in adolescents with first-episode non-suicidal self-injury and the associations with self-injury characteristics"

_PeerJ, doi:10.7717/peerj.19914_

## Round 0.1 · original submission · Major Revisions

Both reviewers have raised major concerns, and the authors are advised to carefully address them.

·

Basic reporting

Language and Clarity: The manuscript is written in clear, professional English. However, there are some areas where phrasing could be improved for greater clarity and fluency.
1. Specific Suggestions:
(1) Lines 77–78: “13 range to 19” could be modified to “range: 13–19 years.” Similarly, Lines 83–84, “14 range to 16; Fig. 1” should be revised to “range: 14–16 years.” Additionally, the reference to “Fig. 1” in this context is unclear; its relevance needs clarification or removal if unnecessary.
(2) Lines 105–106: The phrase “voxel size = 1 mm × 1 mm × 1 mm, slice thickness = 1 mm” appears redundant, as voxel size already encompasses slice thickness. Consider simplifying to “voxel size = 1 mm × 1 mm × 1 mm.”
(3) Lines 112–113: Revise “The total BSSI score is 100, and the higher the score is, the greater the risk of suicide” to “The BSSI has a maximum total score of 100, with higher scores indicating a greater risk of suicide.”
(4) Lines 116–117: Revise “For all the factors, the lower the score for each factor, the worse the cognitive functions” to “For all factors, lower scores indicate poorer cognitive functioning.”
(5) When describing correlations in the Results and Discussion sections, specify the direction (e.g., positive or negative) of the correlations to provide greater clarity and insight. For example: Line 170: “Significant correlation” should be revised to “significant negative correlation.” Add a brief explanatory note, such as: “A greater decrease in GMV is associated with more severe self-injury functions.” Alternatively, summarize these relationships concisely at the end of the relevant paragraph.

Background and Context: no comment.

Figures and Tables: The figures and tables are generally clear and provide relevant information. However, there are areas for improvement to enhance the presentation and interpretability of results:
1. Figure 1: While Figure 1 effectively illustrates the locations of GMV differences using red/blue markers, it could be further improved by displaying the brain atlas contours or shapes instead of just markers. This would provide clearer spatial context for readers. Consider integrating information from Table 3 into Figure 1. For instance, use scatterplots with fitted trend lines to depict significant correlations. This approach would visually highlight the relationships between GMV changes and self-injury functions or cognitive impairments, making the data distribution and results more accessible.
2. Reporting Freedom Degrees: For Tables 1–3, report the degrees of freedom (df) for all statistical tests. Missing participant data (e.g., as mentioned in line 124) should be clearly acknowledged and reflected in these degrees of freedom, ensuring transparency about the sample size used in each analysis.
3. Clarification of Statistical Data: In Table 2, explicitly indicate that the data are represented as “mean ± standard deviation.” This is critical for readers to correctly interpret the presented values.

Data Availability: Raw image data are reported to be available upon request. However, the manuscript would benefit from including a direct link to a publicly accessible data repository or providing a detailed data-sharing statement. This would enhance the transparency and reproducibility of the study, aligning with best practices for open science.

Experimental design

no comment

Validity of the findings

Data Robustness: The statistical methods used are appropriate for the hypotheses tested. The use of covariance analysis and partial correlation is valid but would benefit from justification regarding sample size adequacy for these analyses.
In the statistical analysis section (lines 143–154), it is recommended to include age and sex as covariates. These variables are critical in studies involving structural imaging in adolescents, as both age and sex are known to influence brain structure during this developmental period. Moreover, although no significant differences were found between groups for these variables, their potential effects should be controlled statistically, especially given the relatively small P-values observed.

Interpretation of Results:
The findings are well-linked to the research questions and hypotheses. The relationships among GMV, cognitive impairments, and self-injury functions are comprehensively discussed, providing valuable insights into the neural mechanisms underlying NSSI.
However, while cortical thickness and surface area were mentioned in the methods section as being analyzed, no results or interpretations related to these measures were provided. To enhance transparency, the authors should clarify whether these analyses yielded non-significant findings or were deemed unrelated to the study’s objectives. Including a brief explanation in the results or discussion section would improve the comprehensiveness of the manuscript. Alternatively, these non-significant findings could be provided in supplementary materials for completeness.

Limitations: The authors acknowledge the study’s small sample size and reliance on self-reported data. Including potential biases or confounding variables related to psychosocial factors would strengthen this section.

Additional comments

This study addresses a critical topic with clinical and neuroimaging relevance. Its findings provide valuable insights into the neural correlates of NSSI in adolescents.
1. Suggestions for Additional Analyses:
(1) Mediation Analysis: Explore whether cognitive functions, such as working memory or emotional regulation, mediate the relationship between GMV and self-injury functions. This could reveal the underlying mechanisms connecting brain structure changes to behavioral outcomes.
(2) Interaction Effects: Investigate potential moderators, such as gender or duration of illness, to determine whether these factors influence the relationship between GMV, cognitive impairments, and self-injury functions. Identifying subgroup differences could provide valuable insights for targeted interventions.
(3) Multivariable Analysis: Use multiple regression or structural equation modeling (SEM) to integrate GMV, cognitive functions, and self-injury behaviors into a unified framework. This approach could clarify whether GMV influences self-injury functions directly or indirectly through cognitive impairments, enhancing the understanding of these complex relationships.
2. Suggestions for Discussion:
The manuscript would benefit from a more in-depth discussion of the potential mechanisms linking GMV changes to cognitive impairments and self-injury behaviors. Expanding the discussion to incorporate a broader range of existing neurobiological models could strengthen the theoretical framework and provide valuable context for the findings.
(1) Advanced Imaging Techniques: Consider discussing the potential advantages of 7T T1-Structural scans, which are becoming increasingly accessible. Highlighting how these high-resolution techniques could improve the precision of GMV measurements and lead to more accurate conclusions would add depth to the study. Including such a discussion could also suggest future avenues for research using these advanced technologies.
(2) Additional Imaging Modalities: Incorporate a discussion of complementary imaging modalities, such as functional MRI (fMRI) or diffusion tensor imaging (DTI). These techniques could provide additional insights into brain function and white matter integrity, respectively. By combining these modalities with structural imaging, the study could explore a more comprehensive neural profile of NSSI, potentially uncovering relationships that may not be evident through GMV analysis alone.

·

Basic reporting

• It is unclear if the term “coronavirus” can appropriately be considered a type of pneumonia. You might consider eliminating “pneumonia”.
• The term "NSSI women" might require refinement. Are the authors referring to adults or adolescents with NSSI? The phrasing "women" might not align with contemporary scientific language. I suggest using “assigned female at birth” (AFAB) to ensure inclusivity and accuracy.
• The introduction references various cognitive functions, but the description is quite broad. It would be helpful to clarify which specific cognitive functions are being discussed here.
• Referring to self-injury as a "function" might be misleading. Do the authors mean the underlying mechanisms or maladaptive processes associated with NSSI? A more precise term could enhance clarity.
• The inclusion and exclusion criteria would be better presented prior to listing them in parentheses.

Experimental design

• Regarding imaging methods, I wonder why SPGR was chosen over MPRAGE for gray matter measurement. MPRAGE typically provides better T1-weighted tissue contrast and is widely recognized for more accurate volumetric information. The rationale behind this decision could be elaborated on.
• The choice to use intracranial volume (ICV) as a covariate rather than normalizing gray matter volume (GMV) raises questions. Including the reasoning for this approach would improve transparency.
• Reporting p-values only when they indicate significance is standard practice and would streamline the results section.
• Consistency in p-value formatting (e.g., using "<" or "=" uniformly) would improve readability.
• The description of GM measures is somewhat unclear. The authors mention cortical thickness and surface area but do not indicate whether these measures were significant. Additionally, were all 68 cortical regions analyzed, or were specific regions of interest (ROIs) predefined? These details would improve clarity. Moreover, measures like cortical thickness and surface area also typically require normalization, which does not appear to be addressed in the text.

Validity of the findings

• The manuscript makes some causal inferences that seem unsupported by the statistical methods used. For example, the authors suggest that GMV decreases eventually cause NSSI. However, the reverse (NSSI influencing GMV) is also plausible. Addressing this and discussing potential bidirectional relationships would enhance the paper.
• The comparison made in the manuscript regarding cognitive dysfunction might come across as dismissive of prior work. For example: “Previous research has shown that cognitive dysfunction in NSSI patients is mainly manifested in executive function, working memory, and cognitive processing. By contrast, this study demonstrated that for NSSI adolescents, the impairment of cognitive function includes not only working memory but also overall cognition, motor speed, and verbal fluency.” This phrasing might be reworded to acknowledge the breadth of prior research while emphasizing the unique contributions of the current study. Additionally, cognitive functions represent a vast domain, with various sub-domains, making it difficult to generalize findings without careful specification. The authors might also want to consider checking out recent related work, such as Başgöze et al. (2023), which discusses emotional aspects of cognition, which influences the relationship of NSSI with GM changes:
Başgöze, Z., Demers, L., Thai, M., Falke, C. A., Mueller, B. A., Fiecas, M. B., ... & Cullen, K. R. (2023). A multilevel examination of cognitive control in adolescents with nonsuicidal self-injury. Biological Psychiatry: Global Open Science, 3(4), 855-866.

• The phrase “they cannot make correct decisions” lacks specificity. Further elaboration would be beneficial to clarify what the authors mean here.
• I also noticed that the authors placed significant emphasis on suicide and suicide prevention in the introduction section. However, the discussion section does not specifically address these topics. Expanding on this connection in the discussion might strengthen the paper.

Additional comments

• The language used could benefit from careful review. For example, past studies cannot validate current findings; instead, phrases like "our results supported previous findings" would be more accurate.

---

## Round 0.2 · Major Revisions

There are still several major concerns to address in a revision.

·

Basic reporting

no comment

Experimental design

no comment

Validity of the findings

no comment

Additional comments

no comment

·

Basic reporting

Thank you for your work on this manuscript.

On page 9, the sentence:
“The above results indicate that in the NSSI group, the greater GMV of the left nucleus accumbens or the smaller GMV of the left rACC, the more severe the self-injury function and the worse the cognitive function; moreover, the more severe the self-injury function, the worse the cognitive function.”
The phrase following “moreover” appears redundant, as the preceding part already implies this relationship. If this additional statement is derived from a different statistical analysis, it should be explicitly stated.
In the discussion section, I noticed the use of “alternation” where I believe “alteration” was intended (e.g., on the first page). Also, the page numbers reset to “1” after a certain point.
I would also suggest revising statements that convey high certainty. For instance, rather than saying:
“These alternations are related to cognitive function and the self-injury function.”
You could use a more cautious phrasing, such as:
“Based on these findings, these alterations may be related to cognitive function and self-injury behavior.”
Similarly, the statement:
“The above findings provide imaging and clinical evidence for understanding the pathogenesis of NSSI.”
might be better expressed as:
“These findings contribute to the understanding of the neural mechanisms underlying NSSI.”
For line 367:
“In this study, the cognitive functions of adolescents with NSSI were significantly lower than those of HC.”
It may be clearer to specify:
“The cognitive functions assessed in this study were significantly lower in adolescents with NSSI compared to HC.”

I appreciate your efforts in adjusting terminology regarding sex assignment, such as replacing “women” with “assigned female sex at birth (AFAB).” To maintain consistency, I recommend using “AFAB” instead of “female” throughout the manuscript.

In the following sentence:
“Başgöze et al., through conducting multi-unit analyses on adolescents with NSSI, also found that significant abnormalities in the cognitive functions of NSSI patients were related to NSSI (Başgöze et al., 2023).”
There appears to be a missing word. I believe it should read:
“...were related to NSSI severity.”

Lastly, regarding the phrase “self-injury function,” I still would recommend reconsidering this terminology. The word “function” may imply an inherent or systematic role, similar to “cognitive function,” which could be misleading. Instead, you might consider terms like “neurobehavioral factors associated with self-injurious thoughts and behaviors.”

The discussion section would benefit from careful revision to ensure precise interpretations and to avoid overgeneralizations. Additionally, the language throughout the manuscript should be reviewed for clarity, particularly in the conclusion section. For example:
“...there are abnormal alternations in GMV, including the nucleus accumbens, putamen, and rACC, in adolescents with first-episode NSSI, which are associated with self-injury functions and cognitive functions.”
The phrase “abnormal alternations” is unclear—perhaps “structural variations” or “alterations in GMV” would be more appropriate.

Experimental design

I really appreciate the clarifications regarding MRI data acquisition parameters.
Regarding the normalization of structural data, my previous comment may have been misunderstood. I was referring to the normalization of volume based on each participant’s whole brain volume (i.e., dividing the region volume by whole brain volume) rather than implying the normalization during the preprocessing. If this was not the case in your analysis, then using whole brain volume as a covariate would indeed be an appropriate alternative. My intention was to suggest a way to reduce the number of variables in the model.

The statistics section was somewhat unclear to me. The text refers to a covariance analysis, yet there are figures presenting correlations. Could you clarify whether the term “covariance analysis” refers to ANCOVA?

Validity of the findings

The organization of Figure 2 is difficult to interpret. Is the order of the graphs meaningful? It would be helpful to include more detailed explanations linking these graphs to the statistical analyses described in the methods section. Are these figures intended solely to illustrate relationships, or do they correspond directly to statistical tests? For instance, why is social influence on the x-axis in (d) but on the y-axis in (b)?

Regarding Table 2, it appears to only present significant results. I would recommend displaying all results while highlighting significant ones (e.g., in bold) for a more comprehensive view of the findings.

Additional comments

I appreciate your work on this manuscript and your thoughtful engagement with these revisions. I look forward to seeing the final version.

---

## Round 0.3 · Major Revisions

The second reviewer still raises several major concerns.

·

Basic reporting

The introduction would be strengthened by more clearly articulating the study’s specific hypotheses, particularly regarding 1) Expected differences in GMV and cognitive performance between groups.
2) The theoretical basis for examining mediation (i.e., why GMV is hypothesized to mediate the relationship between NSSI and cognition).

Experimental design

Please clearly state in the methods and abstract that this is a cross-sectional study. While this can be inferred, an explicit statement would help set expectations around causal inference and temporal limitations (especially in the context of the mediation analysis).

The description of the ANCOVA and mediation models could be more precise. For reproducibility and rigor, I recommend the authors 1) confirm whether assumptions for ANCOVA (normality, homogeneity of variance, independence) were tested and met 2) specify the number of bootstrap resamples used in the mediation analysis via PROCESS (e.g., 5000), and report confidence intervals for indirect effects, and 3) clarify whether FDR correction was applied across all voxelwise comparisons or within specific contrasts or regions of interest.

Given that mediation typically implies a causal pathway, the authors should acknowledge more explicitly that the cross-sectional nature of the data limits causal interpretation. This is especially important in the discussion when describing the potential role of GMV in the NSSI-cognition link.

In reporting the main ANCOVA results, please include effect sizes (e.g., partial eta-squared) to help interpret the magnitude of group differences. Also, ensure that the direction of effects is clearly stated throughout (e.g., “reduced GMV in the NSSI group compared to HC”).

Indicate if there was any missing data and how it was handled (e.g., complete case analysis, imputation).

Also, please provide additional detail on how HC participants were screened to ensure they had no psychiatric history, and whether any participants were excluded due to motion artifacts or poor scan quality.

Validity of the findings

It may strengthen the discussion to briefly reflect on how these findings might contribute to early identification or intervention in youth with NSSI.
The limitations section is appropriate but could be slightly expanded to acknowledge potential self-report bias in the assessment of NSSI and any sampling limitations (e.g., gender distribution, treatment status).

Additional comments

Overall, the topic is timely and important, especially given the increasing focus on early identification and intervention in youth with self-injurious behavior. The findings contribute novel insights, particularly through the inclusion of mediation analyses and a well-characterized sample. The authors apparently worked hard to ameliorate their manuscript's language. However, I still believe the manuscript would benefit from clarifying several methodological and reporting elements to enhance transparency and rigor.

---

## Round 0.4 · Minor Revisions

Some minor concerns still exist.

·

Basic reporting

Thank you for addressing the prior concerns. The revised introduction is improved, the authors now clearly articulate their main hypotheses and provide a theoretical rationale for mediation. However, I encourage the authors to further specify their hypotheses by naming the expected brain regions and cognitive domains. Additionally, the justification for mediation could be more strongly framed in terms of timing and mechanism (i.e., why GMV plausibly mediates the NSSI–cognition link). Tightening the narrative and reducing redundancy in the discussion of cognitive function would also enhance clarity.

Experimental design

The authors have addressed all methodological concerns in a satisfactory manner. The cross-sectional design is now clearly stated in both the abstract and methods. Assumptions for ANCOVA are reported, mediation details (e.g., bootstrap resampling, CI reporting) are now clear, and FDR correction is appropriately applied. Missing data were handled via complete case analysis, and HC screening and scan quality control procedures are adequately described. On the other hand, I still recommend a few minor clarifications such as indicating which variables required transformation and rephrasing the motion exclusion sentence. It would also improve clarity to include a CONSORT-style diagram or flow chart summarizing participant inclusion/exclusion and final Ns for each analysis. Overall, the methods section is now much improved and suitable for publication.

Validity of the findings

The results are clearly presented, and the inclusion of effect sizes (partial eta-squared values) enhances the interpretability of group differences. Directionality is now consistently reported, which improves clarity. The discussion appropriately integrates the findings with prior literature and emphasizes the novelty of the observed GMV–cognition relationships. I appreciate the much-ameliorated limitations section and the expanded discussion on early identification. However, some language still implies causality (e.g., “impairs”) which should be further softened to reflect the correlational nature of the data. For clarity and scientific rigor, I encourage the authors to revise remaining phrasing that may overstate causal directionality. The discussion would also benefit from a more cautious interpretation of increased rACC volume, as this finding diverges from prior reports and may require replication.

Additional comments

In terms of presentation, figure captions need rewording for clarity and grammatical correctness. For example, the caption “Greater gray matter volume in adolescents with NSSI than in healthy controls are indicated by red/warm color, and lower volume are indicated by blue/cold color” contains subject-verb agreement errors and is difficult to follow. Some sentences in figure captions feel incomplete or awkwardly structured. Additionally, the third panel in Figure 2 lacks a label for the x-axis, which should be corrected. In Figure 3, the mediation diagram includes paths labeled “c” and “c’” between NSSI and cognitive function, but these are not explained in the caption. Including definitions of c and c’ (i.e., total vs. direct effect) would improve clarity and interpretability.

Overall, the manuscript addresses an important and timely topic and has made significant improvements in response to reviewer feedback. With a few additional clarifications—particularly in tempering causal language, refining figure captions, and addressing minor formatting and labeling issues—the revised manuscript will be substantially strengthened.

Thank you again for your careful and meaningful revisions.

---

## Round 0.5 · accepted · Accept

The authors can be accepted now.

·

Basic reporting

I have no further comments, since the authors have done a great job addressing the issues reported before.

Experimental design

The suggestions are successfully applied.

Validity of the findings

I really appreciate the authors' perseverance and kindness during this review process.